# Immunogenomic engineering of a plug-and-(dis)play hybridoma platform

Mark Pogson[1,*], Cristina Parola[1,2,*], William J. Kelton[1], Paul Heuberger[1] & Sai T. Reddy[1]

Hybridomas, fusions of primary mouse B cells and myelomas, are stable, rapidly-proliferating cell lines widely utilized for antibody screening and production. Antibody specificity of a hybridoma clone is determined by the immunoglobulin sequence of the primary B cell. Here we report a platform for rapid reprogramming of hybridoma antibody specificity by immunogenomic engineering. Here we use CRISPR-Cas9 to generate double-stranded breaks in immunoglobulin loci, enabling deletion of the native variable light chain and replacement of the endogenous variable heavy chain with a fluorescent reporter protein (mRuby). New antibody genes are introduced by Cas9-targeting of mRuby for replacement with a donor construct encoding a light chain and a variable heavy chain, resulting in full-length antibody expression. Since hybridomas surface express and secrete antibodies, reprogrammed cells are isolated using flow cytometry and cell culture supernatant is used for antibody production. Plug-and-(dis)play hybridomas can be reprogrammed with only a single transfection and screening step.

[1] Department of Biosystems Science and Engineering, ETH Zurich, Mattenstrasse 26, Basel 4058, Switzerland. [2] Life Science Zurich Graduate School, Systems Biology, ETH Zurich, University of Zurich, Zurich 8057, Switzerland. * These authors contributed equally to this work. Correspondence and requests for materials should be addressed to S.T.R. (email: sai.reddy@ethz.ch).

Since their inception nearly 40 years ago[1], hybridomas have become one of the most widely utilized platforms for monoclonal antibody (mAb) screening and discovery. Hybridomas are generated by the fusion between primary B cells (typically from immunized mice) and myeloma (plasmacytoma) cells, which results in immortalized, rapidly proliferating stable cultures of antibody producing cell lines, enabling screening, discovery and production of mAbs[2]. By possessing both B cell and plasma cell immunoglobulin RNA splice pathways[3], many hybridoma clones are capable of simultaneously producing both membrane-associated and secretory immunoglobulin heavy (IgH) transcripts, leading to the surface expression and secretion of antibodies[4].

In a typical research lab, the most common approach to recombinant antibody expression is through transient plasmid transfection of mammalian cell lines. Although improvements in plasmid design and delivery has led to systems with high transient expression[5], the constant need to produce and transfect plasmid implies that a stable cell line approach would be advantageous when consistent antibody production is desired. Chinese hamster ovary cells are the predominant stable cell line system for industrial scale production of mAbs, however, hybridomas also have a long history of use in production capacities. This is because the hybridoma fusion partners, myelomas, are derived from plasma cells, which are terminally differentiated B cells possessing a remodelled transcriptional profile and cellular physiology enabling them to secrete large amounts of antibody protein[6]. For example, the plasmacytoma cell lines NS0 and Sp2/0-Ag14 (which do not express endogenous immunoglobulins) have been used extensively for the generation of mAb-producing cell lines, including large-scale manufacturing of several mAb therapeutics[7,8]. However, stable cell line generation relies on random genomic integration of transgenes[9]. Confounding factors such as multiple integration sites, gene silencing, chromatin structure and unbalanced production of antibody heavy and light chains, result in a heterogeneous population where a long and laborious selection process is necessary. This means several months and up to 1 year are typically required before the selection of an optimal stable clone[10]. Therefore, stable cell line generation is typically out of practical reach for academic and small-to-medium-sized entities. A method to reduce the effort and time taken to generate such cell lines by targeted integration of antibody transgenes would be greatly beneficial.

Few examples of targeted genomic modification of hybridomas have been reported. Initially, these studies used hybridomas as model mammalian systems for studying fundamental mechanisms of DNA double-stranded break (DSB) repair[11–13]. In two noteworthy examples, targeted integration at the immunoglobulin locus was used to restore antibody production in an IgG-deficient mutant cell line[14] or for the conversion of the IgH constant region from mouse to human[15]. Although these studies illustrated the potential to genomically modify hybridomas, they relied on classical methods of gene targeting, which tend to be inefficient and require multistep selection systems (for example, neo-HSV-tk)[16]. The emergence of nucleases with programmable targeting specificity, most notably the CRISPR-Cas9 system, has led to a revolution in genome editing applications[17–19]. In a recent example, CRISPR-Cas9 was used to generate DSBs in the immunoglobulin constant region of B cell lines, thereby promoting class-switch recombination or to knock out the IgH constant region for antibody fragment expression[20]. However, to date, the development of a generalizable method capable of exchanging antibody specificity in hybridomas has yet to be described.

Here, we generate a platform for rapid reprogramming of antibody specificity in hybridomas by precise immunogenomic engineering. Our approach is centred on exploiting CRISPR-Cas9 to generate targeted DSBs in the immunoglobulin loci of hybridomas. As a first step, we target the IgH locus and utilized homology directed repair (HDR) to replace the endogenous variable heavy chain ($V_H$) with a donor construct possessing a fluorescent reporter protein, mRuby[21]. Next, we delete the rearranged variable light ($V_L$) chain of the immunoglobulin kappa light (IgK) locus using Cas9 and two guide RNA (gRNA) sites targeting sequences flanking this region. With endogenous antibody expression knocked out, a new antibody coding sequence is introduced into the IgH locus using CRISPR-Cas9 to target and replace mRuby via HDR. The new antibody is encoded by a synthetic antibody region (sAb), which consists of a $V_L$ and constant kappa light chain ($C_k$), a self-cleaving 2A ribosomal skipping peptide, and a $V_H$. This enables expression of full-length IgK and IgH through a single messenger RNA (mRNA) transcript[22], and requires the integration of only a single donor construct into the IgH locus. Since the starting hybridoma clone surface expresses antibody, we are able to isolate cells expressing the sAb using fluorescence-activated cell sorting (FACS), and antibody secretion and specificity is subsequently verified in sorted cells. Finally, we demonstrate the rapid, facile and generalizable nature of this approach by replacing our hybridoma mRuby cell line with several additional sAb constructs. Therefore, we establish a plug-and-(dis)play (PnP) hybridoma platform, where antibody specificity is consistently reprogrammed by CRISPR-Cas9 targeting of the immunoglobulin locus. Using only a single transfection and screening step, we are able to generate cell lines that provide simultaneous antibody surface expression and secretion.

## Results

**Characterization of immunogenomic editing in hybridomas.** We used a parental wild-type (WT) hybridoma cell line with the capacity to surface express and secrete antibodies (clone WEN1.3, derived from prior mouse studies of viral infection, Supplementary Table 1). The IgK and IgH loci were sequenced and CRISPR-Cas9 targeting sites were identified (Supplementary Fig. 1). Protospacer adjacent motif (PAM, 5′-NGG) of S. Pyogenes Cas9 and candidate gRNAs were selected (Supplementary Table 2). For the IgH locus, gRNAs were chosen targeting the intronic region between the leader peptide and the $V_H$ gene and the intronic region immediately downstream of the $J_H$ segment (J4 in $V_H$ of WEN1.3). In the IgK locus, we selected gRNA sites in the leader peptide region and the intronic region downstream of the $J_K$ segment (J4 in $V_L$ of WEN1.3). The gRNAs were cloned into the CRISPR-Cas9 plasmid pX458 (pSpCas9(BB)-2A-GFP), which enables Cas9 expression to be detected via the expression of green fluorescence protein (GFP)[23]. Hybridoma cells were transfected (nucleofection protocol, see the 'Methods' section) with pX458 and flow cytometry revealed a clear population of pX458-positive clones, which were subsequently isolated by FACS (Supplementary Fig. 2a). The effectiveness of Cas9 targeting in the IgH and IgK loci was evaluated using Surveyor assays, which qualitatively measures efficiency of DSB generation as the result of repair through non-homologous end joining (NHEJ)[23]. We found variable levels of Cas9 efficiency for all of the gRNAs sites tested, and in some cases, there was evidence of substantial NHEJ (Supplementary Fig. 2b,c). On the basis of their high activity, for subsequent experiments targeting the immunoglobulin loci of WT cells, we selected gRNA-E for the IgH locus (205 bp downstream of the $J_H$ region), and two gRNAs (gRNA-F, -H) for IgK (targeting the leader region and intron downstream of $J_K$, respectively) (Supplementary Fig. 2b,c).

**Generation of a PnP-mRuby hybridoma cell line.** The first step in generating a PnP hybridoma platform consisted of replacing the $V_H$ region of WT cells with a fluorescent reporter protein using CRISPR-Cas9 targeting (Fig. 1a). An HDR donor construct was designed to include the red fluorescent protein mRuby

flanked by homology arms consistent with the intronic segments flanking the leader and $V_H$ region of WT cells (Fig. 1b and Supplementary Fig. 3). The 5′ homology arm extended 732 bp and terminated immediately before the leader region; the 3′ homology arm started before the end of the J region and

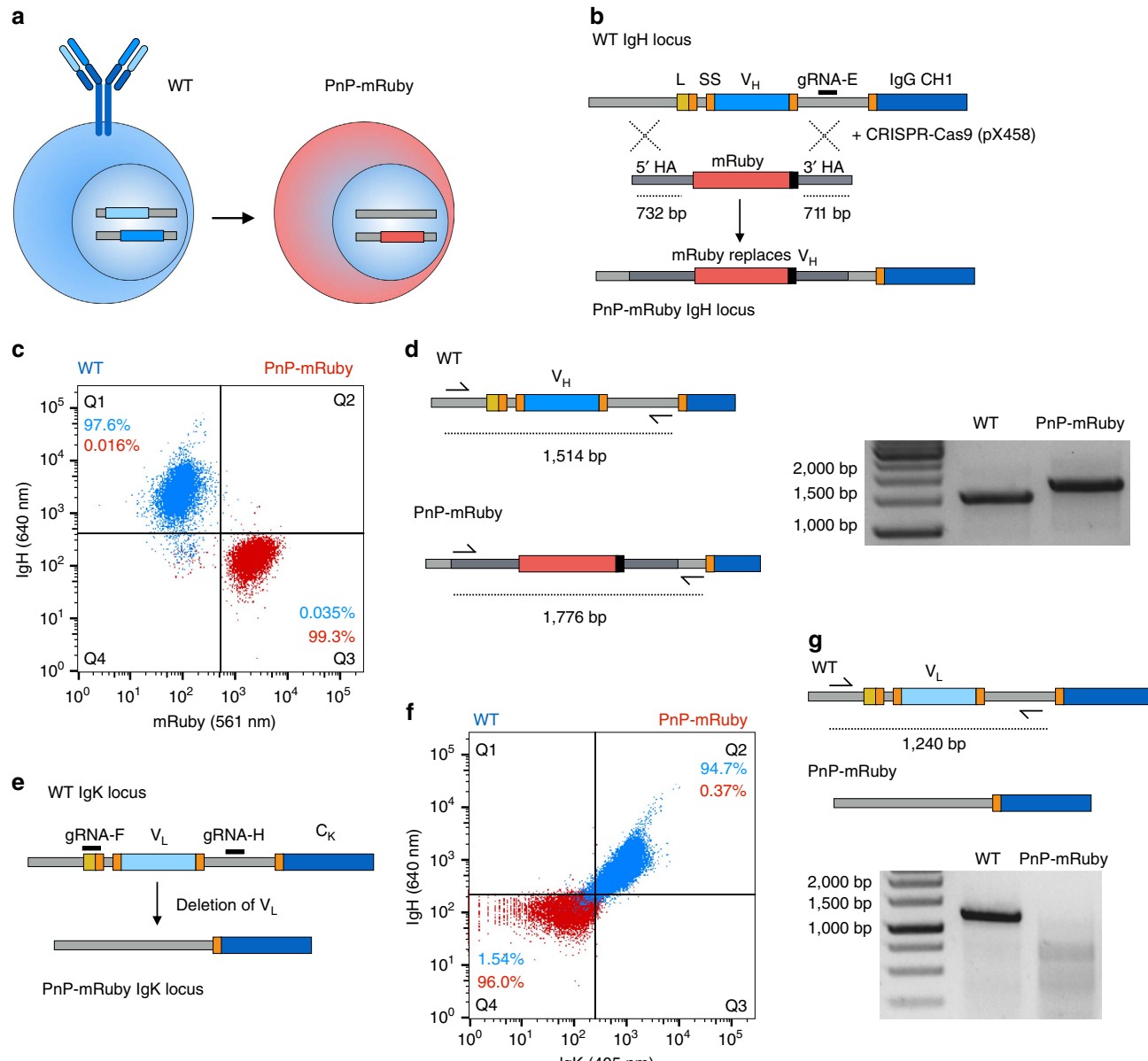

**Figure 1 | Generation of plug-and-(dis)play-mRuby cells. (a)** Schematic shows wild-type (WT) hybridoma cells expressing antibody that will be converted into plug-and-(dis)play (PnP)-mRuby cells. **(b)** Shown is the targeting of WT IgH genomic locus with the following annotations: leader sequence (L, yellow), mRNA splice sites (SS, orange), $V_H$ (blue) and IgG constant heavy region ($C_H1$, dark blue). The Cas9 gRNA-E target site (black) is in the intron between $V_H$ and $C_H1$. The donor construct consists of mRuby gene with a stop codon flanked by two homology arms (HA) of 732 and 711 bp. The PnP-mRuby IgH locus is generated by electroporation of WT cells with CRISPR-Cas9 plasmid and donor construct, which will result in HDR-based exchange of the $V_H$ region with mRuby. **(c)** Flow cytometry dot plot shows that WT cells are exclusively IgH-positive and mRuby-negative, where PnP-mRuby cells are exclusively mRuby-positive and IgH-negative. **(d)** PCR was performed on genomic DNA from WT and PnP-mRuby cells using a forward primer in 5′ HA and reverse primer that is external of the 3′ HA. Agarose gel shows the expected size of bands. The band from PnP-mRuby cells was extracted and Sanger sequencing confirmed mRuby exchange of the $V_H$ region (see Supplementary Fig. 3). **(e)** Shown is the targeting of WT hybridoma IgK locus with the following annotations: $V_L$ (light blue), and IgK constant light region ($C_K$, dark blue), other annotations same as shown in **a**. Two gRNA target sites (black) are utilized to delete the $V_L$ region. **(f)** Flow cytometry dot plot shows WT cells are strongly IgH-positive and IgK-positive, where PnP-mRuby cells are exclusively IgH-negative and IgK-negative. **(g)** PCR was performed on genomic DNA from WT and PnP-mRuby cells using a forward primer 5′ of the gRNA-F site and reverse primer 3′ of gRNA-H site. Agarose gel shows the expected size of band for WT cells and nearly no amplification product for the PnP-mRuby cell line. Throughout this figure, WT cells correspond to clone WEN1.3 and PnP-mRuby cells correspond to clone 1E9.C3 (for more details, see Supplementary Table 1).

extended 711 bp. The WT cells were electroporated with pX458 (with gRNA-E) and the mRuby HDR donor construct (linearized format). At ~48 h post-transfection, Cas9$^+$ cells (2A-GFP$^+$) were isolated by FACS and expanded in culture. At ~10 days post transfection, a clear population of mRuby$^+$ cells was visible; this population was sorted and expanded (Supplementary Fig. 4). As anticipated, mRuby$^+$ cells were predominantly negative for antibody heavy chain expression. Following an additional bulk sort, high-purity single-cell sorting and expansion was performed to obtain a highly pure population (Fig. 1c). To assess whether mRuby integrated site-specifically in the IgH locus, genomic PCR was performed with primers designed to anneal within the 5′ homology arm and downstream (outside) of the 3′ homology arm (Fig. 1d and Supplementary Fig. 4c). The size of PCR products in mRuby$^+$ cells was 1,776 bp (compared with 1,514 bp in WT cells), which corresponded to the expected size for correct integration by HDR. Sanger sequencing of these products confirmed that the mRuby$^+$ gene had indeed replaced the endogenous leader and V$_H$ regions of the IgH locus (Supplementary Fig. 3). The mRuby$^+$ clone 1E9 was selected for the IgK engineering step.

We aimed to make the mRuby$^+$ cell line deficient for light chain expression by deleting the V$_L$ region in the IgK locus (Fig. 1a). In this regard, we took advantage of the facile capacity for multiplexed targeting with CRISPR-Cas9 (ref. 24). We used two gRNAs targeting sites in the leader region (gRNA-F) and 3′ of the J$_K$ segment (gRNA-H) (Fig. 1e). The mRuby$^+$ clone 1E9 was co-transfected with pX458 plasmids encoding both gRNAs and the cells were sorted for Cas9 (2A-GFP) expression and expanded for further characterization. Although replacement of V$_H$ with mRuby should have correspondingly led to an inability for light chain to assemble and express on the surface of cells, we still observed background expression of IgK (Supplementary Fig. 5a). Conveniently, this provided us with a phenotypic selection marker for cells likely to have had their V$_L$ region deleted, thus we used FACS to isolate IgK$^-$ cells. Similar to before, we performed single-cell sorting, expansion and characterization. Several clones demonstrated a clear absence of IgK and IgH surface expression, while maintaining mRuby expression (Fig. 1f). Genomic PCR in the IgK locus confirmed a clear absence of V$_L$, confirming deletion of this region (Fig. 1g and Supplementary Fig. 5b). We selected clone 1E9.C3 for future experiments; for simplicity, this cell line is hereafter referred to as PnP-mRuby (Supplementary Table 1).

**Reprogramming PnP-mRuby cells to express a new antibody.** To reprogram a new antibody specificity into hybridoma cells, PnP-mRuby was used as the starting platform cell line. The presence of a constitutively expressed reporter gene in the IgH locus provided a facile positive–negative phenotype for screening, as mRuby could be replaced with a new antibody and detected by flow cytometry. As before, we first identified and subsequently evaluated gRNA sites (gRNA-J and gRNA-K) targeting the mRuby gene by electroporation of PnP-mRuby cells with pX458.2 (pSpCas9(BB)-2A-BFP, see the 'Methods' section) and Surveyor assays (Supplementary Fig. 6). To reintroduce a new full-length antibody, while avoiding targeting of both the IgK and IgH locus, we used an approach that allowed both full-length light and heavy chain expression from the native IgH promoter as a single transcript (Fig. 2a). This was accomplished by designing a sAb donor template with the following major elements (5′ to 3′): (i) V$_L$ gene (preceded by a leader and intronic segment); (ii) C$_K$ gene; (iii) a self-cleaving 2A peptide combined with a furin cleavage site[22,25]; (iv) V$_H$ gene (preceded by leader and intron region) (Fig. 2b and Supplementary Fig. 7a). The sAb constructs

were cloned into vectors with similar homology arms used for integration of mRuby into the IgH locus (a shortened version of the 3′ homology arm of 697 bp was used), making them compatible for HDR with PnP-mRuby cells. For full-length IgG to express and assemble correctly, mRNA from an integrated V$_H$ gene will have to splice with the first exon of the IgH constant heavy (C$_H$1) region (Fig. 2c and Supplementary Fig. 7b). We included this splice site in the 3′ homology arm of the donor construct, thus providing a phenotypic selection marker of IgH expression for correct sAb insertion. The cells undergoing HDR exchange can also be isolated by positive–negative selection, as mRuby and IgG expression should be mutually exclusive.

We first validated CRISPR-Cas9 targeting and replacement of mRuby with a sAb based on a mAb with specificity for the model antigen hen egg lysozyme (HEL, mAb clone HEL23 derived from plasma cells of an immunized mouse, unpublished results). PnP-mRuby cells were transfected with pX458.2 (gRNA-J) and sAb-HEL23 linear HDR donor. Following sorting and expansion of Cas9$^+$ cells (2A-BFP), we observed a clear population of IgH$^+$ mRuby$^-$ cells (Fig. 2d). Also present were mRuby$^+$ IgH$^-$ and double negative cells. The IgH$^+$ mRuby$^-$ cells were bulk sorted, expanded and extensively characterized (PnP-HEL23). We surface labelled reprogrammed cells for IgK and IgH expression and found them to be almost exclusively positive for IgG surface expression (Fig. 2e). Next, to determine antibody secretion levels, we performed sandwich enzyme-linked immunosorbent assays (ELISAs) on culture supernatants and found PnP-HEL23 expressed IgG, albeit at lower levels compared with the starting WT cells (Fig. 2f). Genomic PCR in the IgH locus (using a forward primer in the 5′ homology arm and reverse primer outside 3′ homology arm) resulted in products of 2,481 bp, the expected size for correct HDR insertion of full-length sAbs comprising V$_L$-C$_K$-2A-V$_H$ (Fig. 2g). PCR with reverse transcription (RT–PCR) on mRNA using a forward primer in the V$_L$ region and reverse primer on C$_H$1 also resulted in a band of the correct size (1,287 bp), suggesting correct splicing of V$_H$ with C$_H$1 transcripts (Fig. 2h). Sanger sequencing on PCR products derived from both genomic DNA and mRNA confirmed sAbs had correctly replaced mRuby in the IgH locus (Supplementary Fig. 7). Finally, we confirmed by flow cytometry and ELISA that both the surface expressed and secreted antibodies from PnP-HEL23 cells retained binding specificity to the HEL antigen (Fig. 3). We also confirmed with independent experiments that transfection of PnP-mRuby cells with pX458.2 containing gRNA-K and a circular sAb plasmid donor (in place of the linearized version), led to HDR and antibody expression (Supplementary Fig. 8).

**Reproducible generation of antibody stable cell lines.** To demonstrate the generalizable nature of our platform, we generated PnP cells using several additional antibodies. We constructed three more sAb donors, keeping the same general format as HEL23. We used variable regions from the following mouse mAbs: HyHEL10, a well-characterized clone with specificity towards HEL[26]; 2G4 and 4G7, antibodies with specificity to Ebola virus (EBV) glycoprotein (murine versions of mAbs used in the therapeutic drug candidate ZMapp)[27]. PnP-hybridomas were electroporated with pX458.2 (gRNA-J) and respective sAb donors, followed by a first screening step for Cas9 expression (2A-BFP). Expansion of sorted cells revealed the presence of mutually exclusive populations expressing IgH or mRuby (Fig. 4a), indicating positive reprogramming with sAb donors. After two sorting steps for IgH$^+$ expression (for PnP-2G4-EBV and PnP-4G7-EBV the second sorting step consisted of double-gating on IgH$^+$ and IgK$^+$ expression), all

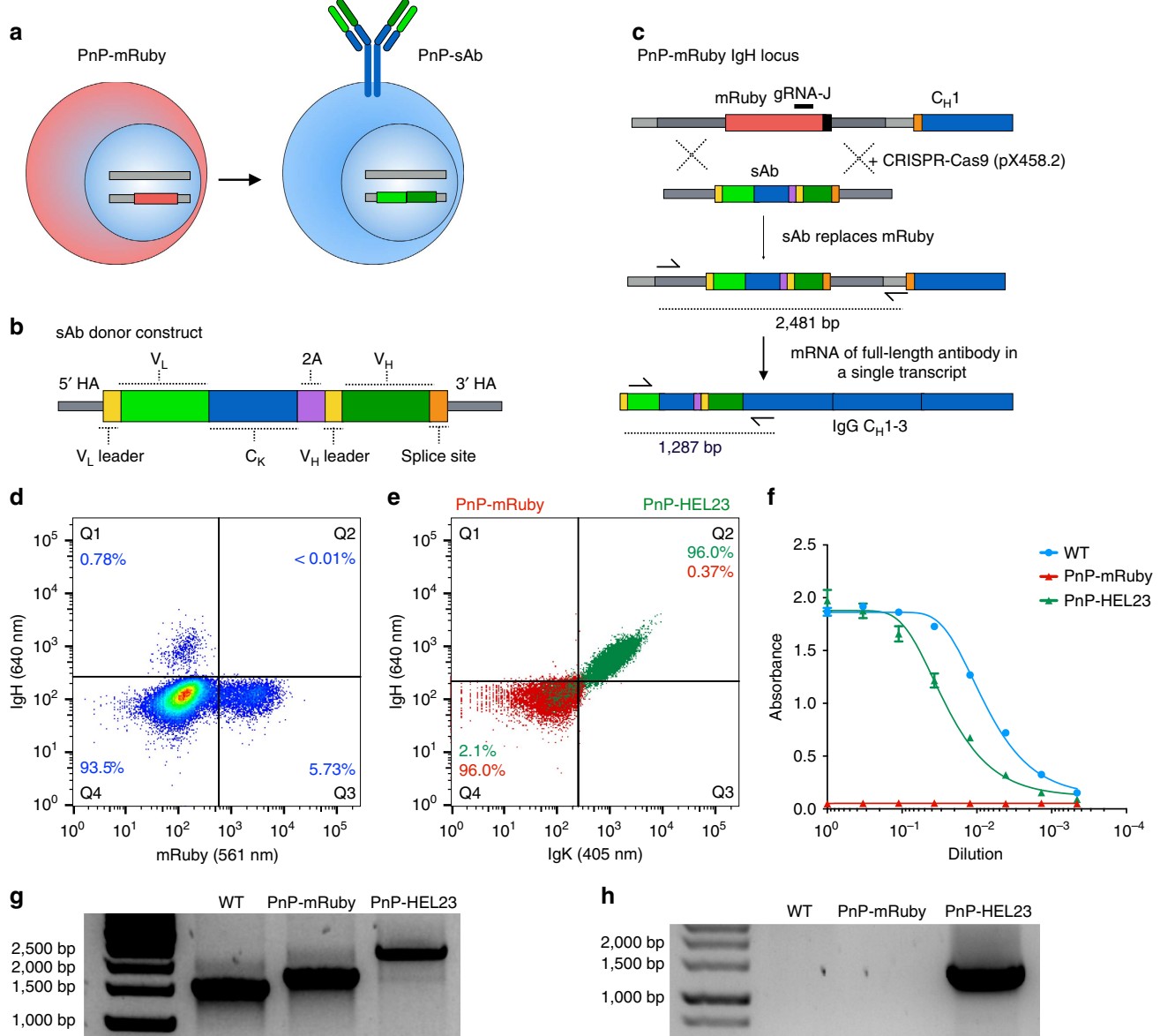

**Figure 2 | PnP-mRuby hybridomas reprogrammed to surface express and secrete a new antibody.** (**a**) Schematic shows PnP-mRuby cells expressing mRuby will be converted back into hybridomas expressing a new antibody via a synthetic antibody (sAb) construct. (**b**) Shown is the design of the sAb donor construct (for simplicity, introns have been excluded, for complete sequence details, see Supplementary Fig. 7). (**c**) Shown is the PnP-mRuby IgH locus where gRNA-J target site (black) is in the mRuby gene. The PnP-mRuby cells transfected with CRISPR-Cas9 plasmid and sAb donor will result in HDR-driven genomic replacement of mRuby. The new antibody will then be expressed from a single mRNA transcript. (**d**) Flow cytometry dot plot shows the different populations that emerge following transfection of PnP-mRuby with CRISPR-Cas9 plasmid and sAb donor [Hen egg lysozyme 23 (HEL23)]. The cells that were positive for IgH expression were sorted. (**e**) Flow cytometry dot plot shows initial population of PnP-mRuby cells and resulting cells (PnP-HEL23) from sorted IgH-positive population in **d**, which are now strongly positive for IgH and IgK expression. (**f**) Graph shows sandwich ELISA results (capture anti-IgK, primary detection anti-IgH) on hybridoma culture supernatant, PnP-HEL23 shows IgG secretion levels similar to WT. For each sample, four technical replicates were analysed and a four-parameter logistical curve was fitted to the data by nonlinear regression. Data are presented as the mean and error bars indicate standard deviation. (**g**) PCR was performed on the genomic DNA of WT, PnP-mRuby, PnP-HEL23 cells using primers shown in **c**. Agarose gel from genomic PCR shows the predicted band size in PnP-HEL23. (**h**) RT–PCR from mRNA was performed with primers shown in **c** and results in a visible band of the correct size present only in PnP-HEL23. The bands from PnP-HEL23 were extracted and Sanger sequencing confirmed correct integration of the PnP-sAb construct (see Supplementary Fig. 7). Throughout this figure, WT cells correspond to clone WEN1.3, PnP-mRuby cells correspond to clone 1E9.C3, and PnP-HEL23 cells correspond to clone Y (for more details, see Supplementary Table 1).

three cell lines were uniformly positive for both IgH and IgK expression (Fig. 4b).

To further demonstrate the simplicity and rapidity of generating new PnP antibody clones, we electroporated PnP-mRuby cells with pX458.2 (gRNA-J) and HEL23 donor, repeating the creation of the PnP-HEL23 (Fig. 2) with the exception of

omitting the initial Cas9 (2A-BFP) sort at ∼48 h post-electroporation. Despite a higher background of mRuby$^+$ cells, a clear IgH$^+$ population was visible (Fig. 4c), with a trend similar to samples subjected to Cas9$^+$ sorting (Fig. 4a). Following sorting for IgH$^+$ expression and expansion, PnP-HEL23.2 cells were uniformly double positive for IgH and IgK expression (Fig. 4d).

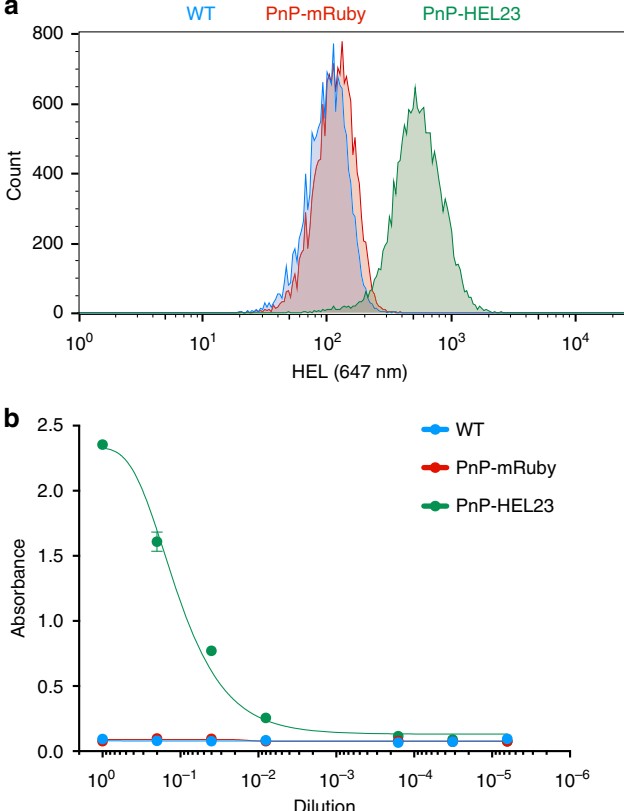

**Figure 3 | Reprogrammed PnP cells surface express and secrete antigen-specific antibody.** (**a**) Flow cytometry histogram shows reprogrammed hybridoma cells (PnP-HEL23) surface express antibody specific for cognate antigen hen egg lysozyme (HEL). (**b**) ELISA data show that PnP-HEL23 cells secrete antibody specific for HEL. One technical replicate was analysed for the control samples (WT and PnP-mRuby) and two replicates for the PnP-HEL23 cell line. A four-parameter logistical curve was fitted to the data by nonlinear regression. Data are presented as the mean and error bars indicate standard deviation. Throughout this figure, WT cells correspond to clone WEN1.3, PnP-mRuby cells correspond to clone 1E9.C3, PnP-HEL23 cells correspond to clone Y (for more details, see Supplementary Table 1).

Sandwich ELISAs for all additional cell lines confirmed there was clear secretion of IgG into the supernatant, although once again at lower levels than parental WEN1.3 cells (Fig. 4e). PCR on mRNA led to a dominant product that indicated correct integration of all sAb donor constructs; however, in the case of PnP-HyHEL10 and PnP-2G4 additional bands were faintly present, possibly indicating alternative insertions of sAb (Fig. 4f, see the 'Discussion' section).

## Discussion

We have established a platform for the immunogenomic reprogramming of hybridoma cells, leading to the rapid generation of cell lines that both surface express and secrete full-length antibodies. Initially, we used Cas9-assisted HDR to replace the $V_H$ exon of the IgH locus with the mRuby gene, creating a phenotypically marked cell line from which to start downstream reprogramming applications. Subsequently, we were able to exchange mRuby with sAb donor cassettes, also via Cas9-assisted HDR. Although it is well established that gene exchange can be accomplished through mechanisms of homologous recombination, this is often reliant on using laborious gene construction technologies to assemble very long

homology arms, which can range from 3 to 5 kb and in some cases up to 20 kb (ref. 28). A more common way to perform gene exchange is through recombinase-mediated cassette exchange (RMCE) using systems such as Cre-loxP or Flp-FRT[29]. However, RMCE requires the *a priori* presence of flanking site-specific recombinase recognition sites ((for example, flanked by loxP (floxed))[30], which would have to be inserted into the genome in advance (via homologous recombination). We in fact initially attempted to use RMCE, using a floxed-mRuby donor (Supplementary Fig. 3). However, after electroporation of PnP-floxed-mRuby cells with Cre recombinase plasmid and floxed donors, we were unable to observe any clear evidence of RMCE. In contrast, when performing mRuby exchange via Cas9-assisted HDR, we were easily able to identify cells having undergone gene exchange (Figs 2 and 4). The simplicity of this method is apparent by the fact that it required only a single electroporation step using a donor cassette with short homology arms (<1 kb). While the current version of our PnP platform uses mouse $C_H$ regions corresponding to IgG2c, in the future Cas9-assisted HDR can be used to replace $C_H$ exons with other Ig subtypes including those from other species such as human[15], rabbit, monkey, which would have value for producing mAb therapeutics and reagents.

Typically, the generation of mAb-producing mammalian cell lines does not result in antibody surface display necessitating the presence of a selectable reporter gene for indicating positive transgene insertion. To eliminate the need for selection based on secondary reporters, antibody surface expression and secretion would be required in the same cells, through alternative splicing (or an inducible switch-based approach)[31]. One such system has been described in mammalian cells through transient transfection of plasmids bearing alternative IgH splicing[32]. An advantage of our PnP platform is that by the use of endogenous hybridoma $C_H$ regions, we are able to leverage native alternative splicing to gain surface expression and secretion of mAbs from the same hybridoma cell. As a result, positive targeted insertion of sAb donors in PnP-mRuby cells is easily detected by surface antibody expression and staining (Figs 2 and 4). Moreover, off-target insertions are not expected to result in full-length antibody transcripts since splicing is necessary between the sAb $V_H$ and the $C_H1$ region in the genome. This alleviates the concern that PnP cells selected for antibody expression were a result of random integration into off-target breaks generated by Cas9 (ref. 33). However, we did in some cases observe a minor presence of larger constructs integrated into the IgH locus in antibody reprogrammed PnP cells (Fig. 4f). It is possible these are a consequence of duplicated sAb donors being inserted into the DSB site via NHEJ[34]. The broader applicability of our PnP system was established by generating multiple cell lines producing several different antibodies. Furthermore, in favour of speed and cost-effectiveness, we were able to optimize the workflow where only a single transfection and screening step was necessary to obtain a homogenous population of antibody producing cells (Fig. 4c,d).

Although our PnP platform enables the rapid and straightforward conversion of hybridoma specificity, it is not without a potential drawback; we observed a reduced level of antibody secretion in all reprogrammed hybridoma cells (Figs 2f and 4e). It is well documented that precise optimization of differential IgK and IgH expression levels leads to more productive recombinant antibody producing stable cells lines[35,36]. In addition, it has been reported that full-length antibody expression using a viral 2A peptide can result in incorrectly processed polypeptides, which attenuates assembly and expression of correctly folded full-length antibodies[37]. In an effort to maintain simplicity by using a single HDR event, the PnP system takes advantage of the native heavy chain promoter for expression of both IgK and IgH as a single

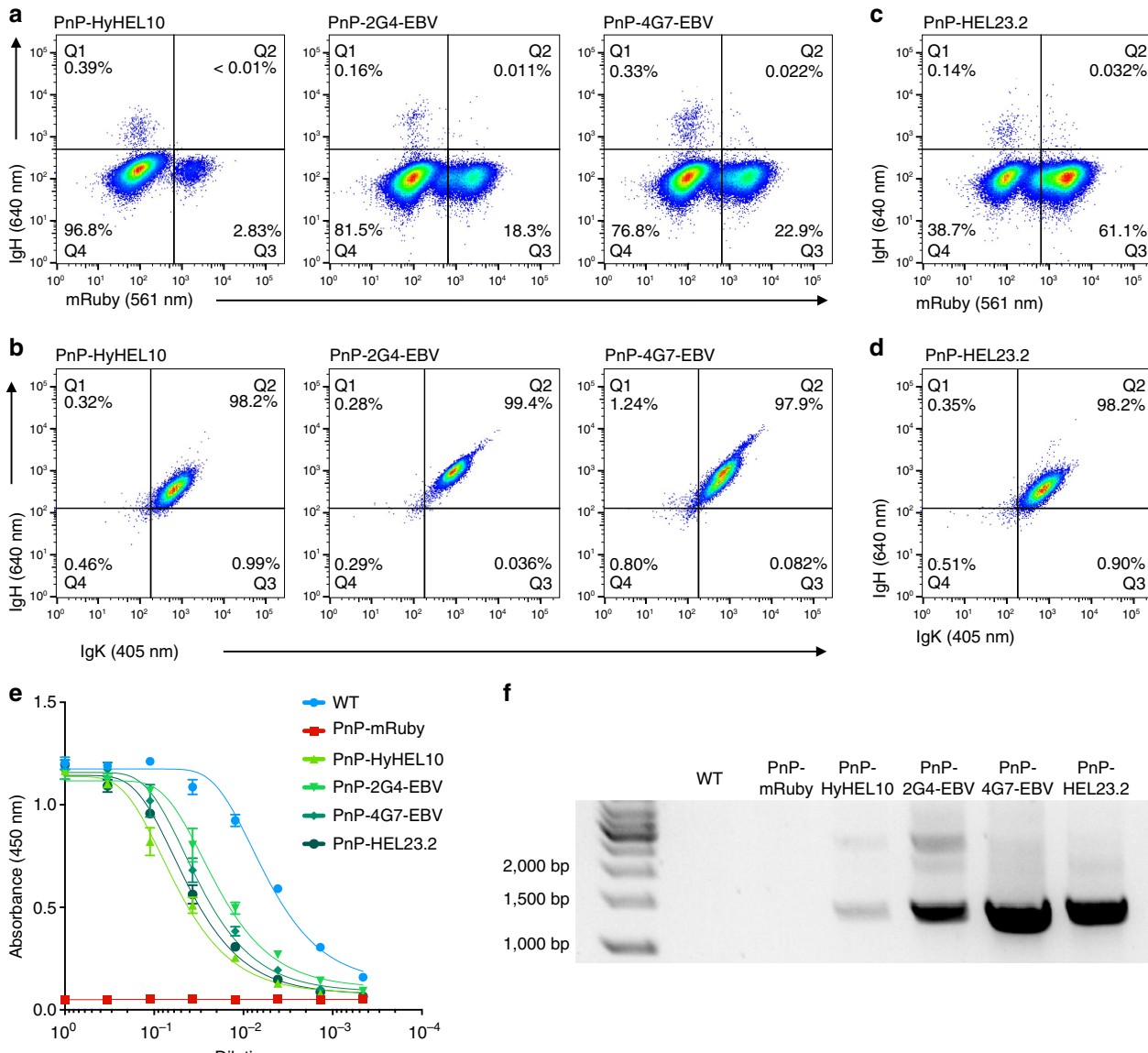

**Figure 4 | Rapid and reproducible antibody reprogramming in PnP-mRuby cells. (a)** Flow cytometry dot plots show PnP-mRuby cells following transfection with CRISPR-Cas9 plasmid and different PnP-sAb donor constructs encoding antibodies specific for HEL or Ebola virus glycoprotein (EBV). The cells were sorted for IgH expression. **(b)** Flow cytometry dot plots show that all three PnP cell lines express IgH and IgK following sorting in **a**. **(c,d)** Similar to **a** and **b** are flow cytometry dot plots for PnP-HEL23.2 cells, which were generated in the same manner as PnP-HEL23 with the exception that the Cas9$^+$ (2A-BFP) sorting step was omitted. **(e)** As in Fig. 2f, plot generated from sandwich ELISA demonstrating that all versions of PnP-antibody producing cells secrete IgG into the supernatant. For each sample, three technical replicates were analysed and a four-parameter logistical curve was fitted to the data by nonlinear regression. Data are presented as the mean and error bars indicate standard deviation. **(f)** As in Fig. 2h, RT–PCR performed on mRNA verified correct insertion and splicing of sAb donors (WT and PnP-mRuby cells received a mixture of forward $V_L$-specific primers). Throughout this figure, PnP-HyHEL10 corresponds to clone U, PnP-2G4-EBV corresponds to clone AA, and PnP-4G7-EBV corresponds to clone AB, PnP-HEL23.2 corresponds to clone AC (for more details, see Supplementary Table 1).

transcript linked by the 2A peptide. Thus, a possible explanation for the observed reduction in secretion is that linked transcripts expressed from a single promoter may result in sub-optimal expression ratios necessary for robust secretion. However, it is interesting to note that although antibody secretion is reduced, we observed similar levels of surface antibody expression in both WT and PnP cells (Figs 1f and 2e). Although reduced antibody secretion is a current limitation of our PnP system, several strategies can be taken in the future to overcome this, such as the use of multiplex-HDR to simultaneously target and exchange $V_L$ and $V_H$ regions, or the inclusion of a second promoter integrated into the sAb donor construct. Furthermore, with an array of

advanced genome editing technologies being developed for transcriptional regulation[38,39], it will be possible to integrate such methods into our PnP system. We envision that by extensively engineering PnP cells, we may eventually be able to achieve mAb production levels comparable to those generated by industrial cell lines.

Even in the presence of a homology donor construct, NHEJ remains the preferred repair mechanism after DSB generation[40], as we indeed saw knockout of mRuby expression in a substantial fraction of cells following transfection with CRISPR-Cas9 (Fig. 2d). However, recent efforts have been made to bias repair towards HDR through the use of both chemical and genetic gene

regulation[41–43]. In addition, it has been shown that constitutive expression of Cas9 in cells is also capable of improving HDR efficiency[44,45]. There is also the potential to optimize gRNA target sites through synthetic protospacers designed to maximize on-target cleavage activity[46]. A combination of these strategies can be applied to our PnP platform and thus may result in greater HDR efficiency, which would further simplify selection. Higher HDR efficiency raises the possibility of eventually incorporating antibody or protein libraries into PnP cells, which could be screened by surface display and thus enable applications in directed evolution and protein engineering[47–53].

A large consortium of biomedical scientists recently reported that a significant investment of research costs and resources are lost due to the use of non-standardized antibody reagents (for example, polyclonal antibodies)[54]. They proposed that a much higher level of reproducibility and consistency would be attained by the standardization of antibody reagents. This first requires greater access to antibody variable region sequences and their associated specificities, which is becoming possible due to the rapid advances in high-throughput sequencing of antibody repertoires[55–57]. In addition, there is also a need to translate the variable region sequences of antibody reagents into recombinant mAbs. This issue highlights the need for systems that enable rapid production of mAb-producing cell lines, which can be conveniently adapted to the needs of a typical research laboratory. By achieving targeted integration, our PnP platform represents a key step forward in making the process more time and cost-efficient, and thus amenable to more researchers. In the context of mAb therapeutics, the 2014 outbreak of Ebola virus infection and the production shortage of the promising drug candidate ZMapp, a cocktail of three mAbs, highlighted the critical need for a technology for faster generation of stable cell lines[58–60]. Although substantial efforts will be needed to improve antibody expression levels, we believe our PnP platform can eventually play a valuable role for therapeutic mAb production as well.

## Methods

**Hybridoma cell culture conditions.** The WT hybridoma cell line (WEN1.3) was obtained as a gift from Professor Annette Oxenius (ETH Zürich). All hybridoma cell lines were cultivated in high-glucose Dulbecco's Modified Eagle Medium ((DMEM), Thermo Fisher Scientific (Thermo), 11960-044) supplemented with 10% heat inactivated fetal bovine serum ((FBS), Thermo, 10082–147), 100 U ml$^{-1}$ penicillin/streptomycin (Thermo, 15140–122), 2 mM glutamine (Sigma-Aldrich, G7513), 10 mM HEPES buffer (Thermo, 15630-056) and 50 µM 2-mercaptoethanol (Sigma-Aldrich, M3148). All hybridoma cells were maintained in incubators at a temperature of 37 °C and 5% CO$_2$. Hybridomas were typically maintained in 10 ml of culture in T-25 flasks (Thermo, NC-156367) and passaged every 48/72 h. The WT hybridoma (WEN1.3) and PnP-mRuby cell line (clone 1E9.C3) were confirmed to be negative for *Mycoplasma* contamination (Universal Mycoplasma Detection Kit, ATCC, 30–1012K).

**Cloning and assembly CRISPR-Cas9 targeting constructs.** The basis for CRISPR-Cas9 experiments relied on the plasmid pSpCas9(BB)-2A-GFP (pX458), obtained as a gift from Feng Zhang (Addgene plasmid #48138)[23]. All gRNAs were obtained from Integrated DNA Technologies (IDT) as single-stranded 5′-phosphorylated oligonucleotides purified by standard desalting. For cloning gRNAs, both versions of pX458 were digested with BbsI (New England BioLabs (NEB), R0539S), gRNA oligonucleotides were ligated into plasmids with DNA T4 ligase (NEB, M0202S). All HDR donor constructs were assembled by Gibson cloning using the Gibson Assembly Master Mix (NEB, E2611S)[61]. When necessary, fragments for the Gibson assembly cloning were amplified with the KAPA HiFi HotStart Ready Mix (KAPA Biosystems (KAPA), KK2602). The gene for mRuby (mRuby2 variant) was derived from the plasmid pcDNA3-mRuby2, a gift from Michael Lin (Addgene plasmid #40260)[62]. The HDR donors (mRuby and the antibody constructs) were cloned in the pUC57(Kan)-HDR plasmid, obtained from Genewiz. The vector was designed with homology arms according to the annotated mouse genomic sequence (GRCm38). The 2A antibody constructs were obtained as synthetic gene fragments (gBlocks, IDT). The mRuby HDR donor was linearized by restriction digestion with the enzyme NruI (NEB, R0192S). The antibody HDR donor vectors were linearized by PCR with the KAPA HiFi HotStart ReadyMix

(KAPA Biosystems, KK2602). An alternate version of pX458 was generated through Gibson cloning by replacing the GFP (eGFP variant) with BPF (TagBFP variant; pX458.2 or pSpCas9(BB)-2A-BFP). All plasmid PX458 and HDR donors were ethanol precipitated as a final purification step.

**Hybridoma transfection with CRISPR-Cas9 constucts.** Hybridoma cells were electroporated with the 4D-Nucleofector System (Lonza) using the SF Cell Line 4D-Nucleofector X Kit L (Lonza, V4XC-2024) with the program CQ-104. The cells were prepared as follows: 10$^6$ cells were isolated and centrifuged at 90g for 5 min, washed with 1 ml of Opti-MEM I Reduced Serum Medium (Thermo, 31985-062) and centrifuged again with the same parameters. The cells were finally re-suspended in 100 µl of total volume of nucleofection mix, containing the vector(s) diluted in SF buffer (per kit manufacturer guidelines). For the exchange of V$_H$ locus, 5 µg of pX458 (or pX458-BFP) with gRNA-E (targeting V$_H$) or gRNA-J (targeting mRuby), and 5 µg of the circular or linearized HDR donor constructs were nucleofected into cells. For V$_L$ deletion, 5 µg each of pX458 with gRNA-F and gRNA-H were co-transfected into cells. Following transfection, the cells were typically cultured in 1 ml of growth media in 24-well plates (Thermo, NC-142475). When a significant cell expansion was observed, the cells were supplemented 24 h later with 0.5–1.0 ml of fresh growth media. After sorting, typically 48 h after transfection, the cells were recovered in 24-well plates, and progressively moved into six-well plates (Thermo, NC-140675) and T-25 flasks, following expansion. After replacing the V$_H$ with mRuby, the cells were single-cell sorted in U-bottom 96-well plates (Sigma-Aldrich, M3562) in a recovery volume of 100 µl. The clones were eventually expanded in 24-well plates, six-well plates and T-25 flasks. The same single-cell sorting procedure was adopted for the isolation of clones after V$_L$ knockout.

**Genomic and transcript analysis of CRISPR-Cas9 targeting.** Genomic DNA of hybridoma cell lines was recovered from typically 10$^6$ cells, which were washed with PBS (Thermo, 10010-015) by centrifugation (250g, 5 min) and re-suspended in QuickExtract DNA Extraction Solution (Epicentre, QE09050). The cells were then incubated at 68 °C for 15 min and 95 °C for 8 min. For transcript analysis, total RNA was isolated from 10$^6$ to 5 × 10$^6$ cells. The cells were lysed with TRIzol reagent (Thermo, 15596-026) and total RNA was extracted with the Direct-zol RNA MiniPrep kit (Zymo Research, R2052). Maxima Reverse Transcriptase (Thermo, EP0742) was used for complementary DNA (cDNA) synthesis from total RNA. Both genomic DNA and cDNA were used as templates for downstream PCR reactions.

The gRNAs targeting WT IgH and IgK loci and mRuby were initially tested for their activity by induction of NHEJ. The targeted fragment was amplified by PCR with KAPA2G Fast ReadyMix (KAPA, KK5121) and the PCR product digested with the Surveyor nuclease for the detection of mismatches (Surveyor Mutation Detection Kit, IDT, 706020). For HDR evaluation, PCR was performed on genomic and cDNA using primers binding inside and outside homology arms (see Supplementary Table 4), followed by fragment size analysis on DNA agarose gels. Selected PCR products were subjected to Sanger sequencing, directly after PCR or following cloning into bacterial plasmids. For the evaluation of HDR exchange in the V$_H$ locus, the following cycling conditions were used on genomic DNA: initial denaturation 3 min at 95 °C; 35 cycles with denaturation at 95 °C (15 s), annealing at 58 °C (15 s), elongation at 72 °C for 1.5 min (PnP-mRuby cell line creation) or 2.6 min (PnP-antibody cell lines creation); final elongation at 72 °C for 1.5 min (PnP-mRuby cell line creation) or 2.6 min (PnP-antibody cell line creation). For the evaluation of V$_L$ deletion, the following cycling conditions were used on genomic DNA: initial denaturation 3 min at 95 °C; 35 cycles with denaturation at 95 °C (15 s), annealing at 61 °C (15 s), elongation at 72 °C for 20 s; final elongation at 72 °C for 1.25 min. For the detection of correctly spliced antibody transcripts, cDNA was amplified with Taq DNA Polymerase with ThermoPol Buffer (NEB, M0267S). The following cycling conditions were used: initial denaturation 3 min at 92 °C; 28 cycles with denaturation at 92 °C (1 min), annealing at 60 °C (1 min), elongation at 72 °C for 1.5 min; final elongation at 72 °C for 7 min. The primers used for amplification and their sequences are listed in Supplementary Table 4. Uncropped versions of the genomic and RT–PCR shown in Figs 1, 2 and 4 are reported in Supplementary Fig. 9. For reference, the DNA size markers used are GeneRuler 1 kb DNA Ladder (Thermo, SM0314) and GeneRuler 100 bp DNA Ladder (Thermo, SM0243).

**Flow cytometry analysis and sorting of hybridomas.** Flow cytometry-based analysis and cell isolation were performed using the BD LSR Fortessa and BD FACS Aria III (BD Biosciences), respectively. At 24 h post transfection, ~100 µl of cells were collected, centrifuged at 250g for 5 min, resuspended in PBS and analysed for expression of Cas9 (via 2A-GFP/-BFP). Forty-eight hours post transfection, all the transfected cells were collected and resuspended in Sorting Buffer (SB (PBS supplemented with 2 mM EDTA and 0.1% BSA)). When labelling was required, the cells were washed with PBS, incubated with the labelling antibody or antigen for 30 min on ice, protected from light, washed again with PBS and analysed or sorted. The labelling reagents and working concentrations are described in Supplementary Table 3. For cell numbers different from 10$^6$, the antibody/antigen amount and incubation volume were adjusted proportionally.

**Measurement of antibody secretion by ELISA.** Sandwich ELISAs were used to measure the secretion of IgG from hybridoma cell lines. The plates were coated with capture polyclonal anti-light chain (goat anti-mouse, Jackson ImmunoResearch, 115-005-174) concentrated at 3.7–4 µg ml$^{-1}$ (1:325 dilution from stock) in PBS (Thermo, 10010-015). The plates were then blocked with PBS supplemented with 2% m/v milk (AppliChem, A0830) and 0.05% V/V Tween-20 (AppliChem, A1389, PBSMT). After blocking, plates underwent three washing steps with PBS supplemented with Tween-20 0.05% V/V (PBST). Supernatants from cell culture (10$^6$ cells per sample, volume normalized to least-concentrated samples) were then serially diluted (at 1:3 ratio) in PBS supplemented with 2% m/v milk (PBSM), starting from the non-diluted supernatant as the highest concentration. Supernatants were incubated for 1 h at room temperature or overnight at 4 °C, followed by three washing steps with PBS supplemented with Tween-20 0.05% V/V (PBST). HRP-conjugated polyclonal anti-mouse Fc region (goat anti-mouse, Sigma-Aldrich, A2554) was used as secondary antibody, concentrated at 1.7 µg ml$^{-1}$ (1:5,000 dilution from stock) in PBSM, followed by three washing steps with PBST. ELISA detection was performed using a 1-Step Ultra TMB-ELISA Substrate Solution (Thermo, 34028) as the HRP substrate, reaction was terminated with H$_2$SO$_4$ (1 M). Absorbance at 450 nm was read with Infinite 200 PRO NanoQuant (Tecan).

For antigen-specificity measurements, the plates were coated with purified hen egg lysozyme (Sigma-Aldrich, 62971-10G-F) concentrated at 4 µg ml$^{-1}$ in PBS. Plates were then blocked with PBS supplemented with 2% m/v milk (AppliChem, A0830) and 0.05% V/V Tween-20 (AppliChem, A1389) (PBSMT). After blocking, the plates underwent three washing steps with PBS supplemented with Tween-20 0.05% V/V (PBST). Supernatants from cell culture (10$^6$ cells per sample, volume normalized to least concentrated samples) were then serially diluted (at 1:5 ratio) in PBS supplemented with 2% m/v milk (PBSM), starting from the non-diluted supernatant as the highest concentration. A HRP-conjugated monoclonal anti-mouse kappa light chain (rat anti-mouse, Abcam, AB99617) concentrated at 0.7 µg ml$^{-1}$ (1:1,500 dilution from stock). ELISA detection was performed using a 1-Step Ultra TMB-ELISA Substrate Solution (Thermo, 34028) as the HRP substrate, reaction was terminated with H$_2$SO$_4$ (1 M). Absorbance at 450 nm was read with Infinite 200 PRO NanoQuant (Tecan). ELISA data were analysed with the software GraphPad Prism.

**Data availability.** All the cell lines, materials and relevant data generated in this study are available by request. Relevant genomic DNA and/or cDNA sequences from the WT hybridoma, PnP-mRuby, PnP-HEL23 cell lines and HDR donor vectors are available on GenBank (http://www.ncbi.nlm.nih.gov/genbank/) with the following accession numbers: KX398103 (Wen1.3 hybridoma IgH genomic locus), KX431572 (Wen1.3 hybridoma IgK genomic locus), KX377895 (PnP-mRuby edited engineered IgH genomic locus), KX431573 (PnP-HEL23 edited VH locus from genome), KX431574 (PnP-HEL23 spliced synthetic antibody (sAb)), KX431576 (pUC57(Kan)-mRuby2-HDR homology donor plasmid), KX431575 (pUC57(Kan)-HEL23-HDR homology donor plasmid).

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

## Acknowledgements

We acknowledge ETH Zurich D-BSSE Single Cell Unit for support, in particular, M. Dessing, T. Lopes, V. Jäggin. We also thank N. Mehta, A. Renz and D. Mason for assistance with molecular cloning experiments and T. Khan for scientific discussions. This work was supported by the ETH Zurich Postdoctoral Fellowship (to M.P.); Swiss National Science Foundation (to S.T.R.); The National Center of Competence in Research (NCCR) Molecular Systems Engineering (to S.T.R.). The professorship of S.T.R. is supported by an endowment from the S. Leslie Misrock Foundation.

## Author contributions

M.P., C.P. and S.T.R. developed the methodology and wrote the manuscript; M.P., C.P., W.J.K. and S.T.R. designed the experiments. M.P., C.P. and P.H. performed the experiments.

## Additional information

**Competing financial interests:** ETH Zurich has filed for patent protection on the technology described herein, and M.P., C.P., W.J.K. and S.T.R. are named as co-inventors on this patent (European Patent Application: 16163734.3-1402).

