## [Peer review file · Nature Communications]

Reviewers' Comments:

Reviewer #1 (Remarks to the Author)

In their manuscript, Pogson et al., offer a simple yet elegant way to change the antigen specificity of murine B cell hybridomas. They successfully employed the CRISPR/Cas9 system to delete the endogenous variable light chain and replaced the variable heavy chain with a cassette encoding the fluorescent reporter gene mRuby. The resulting cell product represents a platform that can easily incorporate new antibody genes, in a plug-and-(dis)play fashion, under control of the endogenous heavy chain promoter. A great advantage of this system, over existing technologies, such as the generation of stable cell lines by random integration, is the fact that the inserted antibody will not only be secreted but also be produced in its membrane-bound form, allowing the facile isolation of positive clones by FACS sorting. The authors demonstrate the ease of using their platform by generating 4 different hybridomas secreting antibodies with specificity to the model antigen HEL and the clinically relevant Ebola virus glycoprotein.

One caveat of their system seems to be that the overall levels of secreted antibody appear to be lower than in the original hybridoma line, and are not at par with levels required for industrial scale production. Future directions, such as improving expression levels, enhancing recombination efficiencies, or replacing the murine constant region with a human version for the production of humanized antibodies are appropriately discussed and beyond the scope of this manuscript. While the authors are not the first to apply the CRISPR/Cas9 system to murine hybridoma cell lines (see Cheong et al., Nat Communications 2016), the PnP platform described in this study is certainly novel and, given its potential applications in biotechnology and in basic research, of great interest to a large readership.

Minor comments:

- The introduction can be significantly shortened if space is an issue. E.g. the last paragraph of the introduction can be integrated into the results section.

- The figure legends are very technical, and may require a facelift. I would not include the plasmid names (pX458 or pX458.2) in the figures/figure legends. Why not name the guide RNA used in each experiment in the figure legend, e.g. 'CRISPR-Cas9 plasmid containing gRNA-XYZ'? Moreover, as a reader I'd appreciate spelling out non-standard abbreviations when used in figures for the first time, even though the authors did indeed introduce them in the main text. PnP in Fig.1, sAB in Fig.2, HEL in Fig.3. It also would not hurt to include the antigen specificities (hen egg lysozyme and Ebola virus glycoprotein) in figure legend 4., instead of just naming the non-descriptive clone names.

Given the broad implications of their PnP system for biotechnology and the larger research community, I recommend to accept the manuscript with minor revisions.

Reviewer #2 (Remarks to the Author)

This is essentially a methods paper. Sufficient detail is provided for others to carry out the methods. The paper is clearly written. Although unlikely to have a major impact on the field, the paper is a novel application of the burgeoning CRISPR-Cas9 technology.

The authors have attempted to develop a novel platform for expressing antibodies of different specificities in mouse hybridoma cells. To do so they have used CRISPR-Cas9 technology to delete VL expression and also to introduce the fluorescent report protein mRuby into the VH locus. CRISPR-Cas9 technology can then be used to introduce a novel synthetic antibody comprised of light chain (V and C regions) a 2A ribosomal skipping peptide, and a VH with the novel synthetic

antibody replacing mRuby. Cells of the desired phenotype are mRuby⁻, H⁺ and can be selected using flow cytometry. Their claim is that this approach is more rapid and efficient than current methods of either transient transfection or selection of stable transfectants. That claim is probably true if they make the PnP-mRuby line freely available to investigators and the investigators have access to a facility that will perform the necessary cell sorting.

However, it should be noted that the approach has some limitations. As described, all of the antibodies will have the mouse IgG2c heavy chain constant region, thereby limiting the number of potential applications. The method can only be used if the sequences of the VL and VH to be expressed are known; in fairness this limitation applies to other methods of gene transfection. However what is most disappointing is the expression level of the resulting clones. Although some lines appear to express at about 50% of the level of the original hybridoma, others appear to make only about 10% of that amount. One of the clear motivations of using this approach is to be able to achieve hybridoma like expression levels by targeting the endogenous locus; clearly that expectation is not reached. An unanswered question is why; is it the structure of the inserted fusion gene, or some other problem? The fusion gene is designed to make equal amounts of H and L chains and in some cases excess L chain is required to facilitate efficient antibody secretion.

Immunogenomic engineering of a plug-and-(dis)play hybridoma platform

Point-by-Point Response to Reviewers

Reviewer 1 (Remarks to Author):

R1.1) *In their manuscript, Pogson et al., offer a simple yet elegant way to change the antigen specificity of murine B cell hybridomas. They successfully employed the CRISPR/Cas9 system to delete the endogenous variable light chain and replaced the variable heavy chain with a cassette encoding the fluorescent reporter gene mRuby. The resulting cell product represents a platform that can easily incorporate new antibody genes, in a plug-and-(dis)play fashion, under control of the endogenous heavy chain promoter. A great advantage of this system, over existing technologies, such as the generation of stable cell lines by random integration, is the fact that the inserted antibody will not only be secreted but also be produced in its membrane-bound form, allowing the facile isolation of positive clones by FACS sorting. The authors demonstrate the ease of using their platform by generating 4 different hybridomas secreting antibodies with specificity to the model antigen HEL and the clinically relevant Ebola virus glycoprotein.*

We thank the reviewer very much for their insightful and positive comments.

R1.2) *One caveat of their system seems to be that the overall levels of secreted antibody appear to be lower than in the original hybridoma line, and are not at par with levels required for industrial scale production.*

The reviewer makes a very good point that expression levels of secreted antibodies in reprogrammed hybridomas were lower than that of the original hybridoma clone. We have extensively addressed this point in the discussion of the manuscript and also addressed a similar point raised by Reviewer 2.

Please see below response to R2.5.

See revised Discussion: lines 273-293 and refs. 35-37.

R1.3) *Future directions, such as improving expression levels, enhancing recombination efficiencies, or replacing the murine constant region with a human version for the production of humanized antibodies are appropriately discussed and beyond the scope of this manuscript. While the authors are not the first to apply the CRISPR/Cas9 system to murine hybridoma cell lines (see Cheong et al., Nat Communications 2016), the PnP platform described in this study is certainly novel and, given its potential applications in biotechnology and in basic research, of great interest to a large readership.*

We thank the reviewer for the positive comments and acknowledging the future potential of our PnP platform.

Minor Comments

R1.4) *The introduction can be significantly shortened if space is an issue. E.g. the last paragraph of the introduction can be integrated into the results section.*

We appreciate the reviewer's concern regarding the length of the last paragraph of the introduction, however currently space is not a limitation as the introduction is well below the 1,000-word limit (841 words). We would like to keep this final introduction paragraph as is because we introduce important terminology used throughout the manuscript. We believe keeping this in the introduction will make it easier for readers to follow the results section.

R1.5) *The figure legends are very technical, and may require a facelift. I would not include the plasmid names (pX458 or pX458.2) in the figures/figure legends. Why not name the guide RNA used in each*

experiment in the figure legend, e.g. 'CRISPR-Cas9 plasmid containing gRNA-XYZ'? Moreover, as a reader I'd appreciate spelling out non-standard abbreviations when used in figures for the first time, even though the authors did indeed introduce them in the main text. PnP in Fig.1, sAB in Fig.2, HEL in Fig.3. It also would not hurt to include the antigen specificities (hen egg lysozyme and Ebola virus glycoprotein) in figure legend 4., instead of just naming the non-descriptive clone names.

We thank the reviewer for their comments regarding some of the technical terminology used in the figure legends. We have gone through each figure legend and accordingly made changes suggested by the reviewer, we believe the legends should now be easier for a reader to understand.

R1.6) *Given the broad implications of their PnP system for biotechnology and the larger research community, I recommend to accept the manuscript with minor revisions.*

We thank the reviewer for providing such a strong overall assessment of the manuscript.

Reviewer 2 (Remarks to Author):

R2.1) *This is essentially a methods paper. Sufficient detail is provided for others to carry out the methods. The paper is clearly written. Although unlikely to have a major impact on the field, the paper is a novel application of the burgeoning CRISPR-Cas9 technology. The authors have attempted to develop a novel platform for expressing antibodies of different specificities in mouse hybridoma cells. To do so they have used CRISPR-Cas9 technology to delete VL expression and also to introduce the fluorescent report protein mRuby into the VH locus. CRISPR-Cas9 technology can then we used to introduce a novel synthetic antibody comprised of light chain (V and C regions) a 2A ribosomal skipping peptide, and a VH with the novel synthetic antibody replacing mRuby. Cells of the desired phenotype are mRuby-, H+ and can be selected using flow cytometry.*

We thank the reviewer for their overall summary of the manuscript, and for highlighting the novelty of our PnP platform and the detail provided that allows future researchers to use the platform.

R2.2) *Their claim is that this approach is more rapid and efficient than current methods of either transient transfection or selection of stable transfectants. That claim is probably true if they make the PnP-mRuby line freely available to investigators and the investigators have access to a facility that will perform the necessary cell sorting.*

We are grateful that the reviewer agrees that our PnP platform is more rapid and efficient than the current standard for recombinant antibody production based on transient transfection or stable cell line generation by random integration. This does indeed suggest that many investigators will be interested in accessing our platform. We will certainly make PnP cells, materials, and sequences generated in this study available to interested researchers. We have added a statement explicitly stating this in the methods section.

See revised Methods Data Availability: lines 446-450.

R2.3) *However, it should be noted that the approach has some limitations. As described, all of the antibodies will have the mouse IgG2c heavy chain constant region, thereby limiting the number of potential applications.*

The reviewer is correct that the current version of our PnP platform is limited to the mouse IgG2c constant region domain. We have shown in this study that we were able to use CRISPR-Cas9 and HDR to replace the variable heavy chain region in hybridomas. This approach can be readily adapted to also replace the constant IgG2c heavy chain region with other Ig subtypes or that of other species such as human. We have now more explicitly stated in the discussion the ability to further engineer constant regions by CRISPR-Cas9 and HDR.

See revised Discussion: lines 246-250.

R2.4) *The method can only be used if the sequences of the VL and VH to be expressed are known; in fairness this limitation applies to other methods of gene transfection.*

The reviewer makes a very valid point that to reprogram hybridomas requires *a priori* knowledge of the VL and VH genes. However, a number of advances have recently been made in obtaining greater sequencing information on variable regions by performing high-throughput sequencing on antibody repertoires. We believe these advances can synergize with our PnP platform to solve one of the major research challenges that exist in the field – antibody reagent standardization. We have revised the discussion to highlight this point with more clarity and added additional references.

See revised Discussion: lines 310-314 and refs. 55-57.

R2.5) *However what is most disappointing is the expression level of the resulting clones. Although some lines appear to express at about 50% of the level of the original hybridoma, others appear to make only about 10% of that amount. One of the clear motivations of using this approach is to be able to achieve hybridoma like expression levels by targeting the endogenous locus; clearly that expectation is not reached. An unanswered question is why; is it the structure of the inserted fusion gene, or some other problem? The fusion gene is designed to make equal amounts of H and L chains and in some cases excess L chain is required to facilitate efficient antibody secretion.*

The reviewer has very correctly pointed out that a limitation of our PnP system is related to the reduced level of antibody secretion in reprogrammed cells when compared to that of the parental hybridoma used in this study. We have identified several potential reasons this reduced antibody production occurs, they are the following:

1. PnP cells rely on IgK and IgH expression all from a single promoter whereas WT cells rely on the use of two separate promoters for independent expression of IgK and IgH.
2. As correctly pointed out by the Reviewer, the differential ratio of IgK and IgH results in more optimal antibody expression levels (see refs. 35-36). WT cells may have a more optimal ratio than PnP cells, once again due to the presence of two versus one promoter.
3. Unlike WT cells, PnP cells rely on the self-cleaving 2A peptide, it has been observed that full-length antibodies using this system can result in polypeptide processing defects. This results in a lower level of proper antibody folding, assembly, and expression (see refs. 37).

We have now revised the Discussion section of the manuscript by adding an entire new paragraph that documents the four above mentioned reasons for reduced antibody expression in PnP cells along with associated references. Furthermore, we have added very specific strategies that can be readily implemented into our PnP system to improve antibody expression, as well as long-term approaches that rely on novel genome editing technologies.

See revised Discussion: lines 273-293 and new refs. 35-37.

Reviewers' Comments:

Reviewer #1 (Remarks to the Author)

The authors addressed all comments to my full satisfaction. I recommend the manuscript for publishing.

Reviewer #2 (Remarks to the Author)

There are some very small issues:

"data" is a plural word and in several places was used as if it were singular.

In several places the references to the figures used a capital letter, where it should always be lower case to be consistent.